# CPR: Classifier-Projection Regularization for Continual Learning

**Sungmin Cha**[1], **Hsiang Hsu**[2], **Taebaek Hwang**[1], **Flavio P. Calmon**[2], **and Taesup Moon**[3]*
[1]Sungkyunkwan University     [2]Harvard University     [3]Seoul National University
csm9493@skku.edu, hsianghsu@g.harvard.edu, gxq9106@gmail.com,
fcalmon@g.harvard.edu, tsmoon@snu.ac.kr

## ABSTRACT

We propose a general, yet simple patch that can be applied to existing regularization-based continual learning methods called classifier-projection regularization (CPR). Inspired by both recent results on neural networks with wide local minima and information theory, CPR adds an additional regularization term that maximizes the entropy of a classifier's output probability. We demonstrate that this additional term can be interpreted as a projection of the conditional probability given by a classifier's output to the uniform distribution. By applying the Pythagorean theorem for KL divergence, we then prove that this projection may (in theory) improve the performance of continual learning methods. In our extensive experimental results, we apply CPR to several state-of-the-art regularization-based continual learning methods and benchmark performance on popular image recognition datasets. Our results demonstrate that CPR indeed promotes a wide local minima and significantly improves both accuracy and plasticity while simultaneously mitigating the catastrophic forgetting of baseline continual learning methods. The codes and scripts for this work are available at https://github.com/csm9493/CPR_CL.

## 1 INTRODUCTION

Catastrophic forgetting (McCloskey & Cohen, 1989) is a central challenge in continual learning (CL): when training a model on a new task, there may be a loss of performance (e.g., decrease in accuracy) when applying the updated model to previous tasks. At the heart of catastrophic forgetting is the *stability-plasticity dilemma* (Carpenter & Grossberg, 1987; Mermillod et al., 2013), where a model exhibits high stability on previously trained tasks, but suffers from low plasticity for the integration of new knowledge (and vice-versa). Attempts to overcome this challenge in neural network-based CL can be grouped into three main strategies: regularization methods (Li & Hoiem, 2017; Kirkpatrick et al., 2017; Zenke et al., 2017; Nguyen et al., 2018; Ahn et al., 2019; Aljundi et al., 2019), memory replay (Lopez-Paz & Ranzato, 2017; Shin et al., 2017; Rebuffi et al., 2017; Kemker & Kanan, 2018), and dynamic network architecture (Rusu et al., 2016; Yoon et al., 2018; Golkar et al., 2019). In particular, regularization methods that control model weights bear the longest history due to its simplicity and efficiency to control the trade-off for a fixed model capacity.

In parallel, several recent methods seek to improve the generalization of neural network models trained on a single task by promoting *wide local minima* (Keskar et al., 2017; Chaudhari et al., 2019; Pereyra et al., 2017; Zhang et al., 2018). Broadly speaking, these efforts have experimentally shown that models trained with wide local minima-promoting regularizers achieve better generalization and higher accuracy (Keskar et al., 2017; Pereyra et al., 2017; Chaudhari et al., 2019; Zhang et al., 2018), and can be more robust to weight perturbations (Zhang et al., 2018) when compared to usual training methods. Despite the promising results, methods that promote wide local minima have yet to be applied to CL.

In this paper, we make a novel connection between wide local minima in neural networks and regularization-based CL methods. The typical regularization-based CL aims to preserve *important* weight parameters used in past tasks by penalizing large deviations when learning new tasks. As

---

*Corresponding author (E-mail: tsmoon@snu.ac.kr)

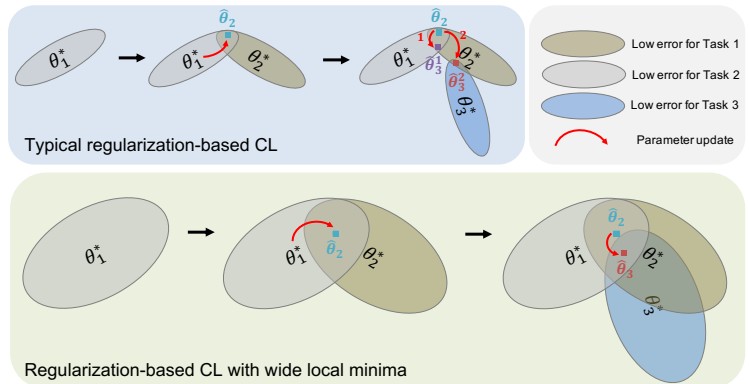

Figure 1: In typical regularization-based CL (top), when the low-error ellipsoid around local minima is sharp and narrow, the space for candidate model parameters that perform well on all tasks (*i.e.*, the intersection of the ellipsoid for each task) quickly becomes very small as learning continues, thus, an inevitable trade-off between stability and plasiticty occurs. In contrast, when the *wide local minima* exists for each task (bottom), it is more likely the ellipsoids will significantly overlap even when the learning continues, hence, finding a well performing model for all tasks becomes more feasible.

shown in the top of Fig. 1, a popular geometric intuition (as first given in EWC (Kirkpatrick et al., 2017)) for such CL methods is to consider the (uncertainty) ellipsoid of parameters around the local minima. When learning new tasks, parameter updates are selected in order to not significantly hinder model performance on past tasks. Our intuition is that promoting a wide local minima—which conceptually stands for local minima having a *flat*, rounded uncertainty ellipsoid—can be particularly beneficial for regularization-based CL methods by facilitating diverse update directions for the new tasks (*i.e.*, improves plasticity) while not hurting the past tasks (*i.e.*, retains stability). As shown in the bottom of Fig. 1, when the ellipsoid containing the parameters with low-error is wider, *i.e.*, when the wide local minima exists, there is more flexibility in finding a parameter that performs well for all tasks after learning a sequence of new tasks. We provide further details in Section 2.1.

Based on the above intuition, we propose a general, yet simple patch that can be applied to existing regularization-based CL methods dubbed as *Classifier-Projection Regularization* (CPR). Our method implements an additional regularization term that promotes wide local minima by maximizing the entropy of the classifier's output distribution. Furthermore, from a theory standpoint, we make an observation that our CPR term can be further interpreted in terms of information projection (I-projection) formulations (Cover & Thomas, 2012; Murphy, 2012; Csiszár & Matus, 2003; Walsh & Regalia, 2010; Amari et al., 2001; Csiszár & Matus, 2003; Csiszár & Shields, 2004) found in information theory. Namely, we argue that applying CPR corresponds to projecting a classifier's output onto a Kullback-Leibler (KL) divergence ball of finite radius centered around the uniform distribution. By applying the Pythagorean theorem for KL divergence, we then prove that this projection may (in theory) improve the performance of continual learning methods.

Through extensive experiments on several benchmark datasets, we demonstrate that applying CPR can significantly improve the performance of the state-of-the-art regularization-based CL: using our simple patch improves *both* the stability and plasticity and, hence, achieves better average accuracy almost uniformly across the tested algorithms and datasets—confirming our intuition of wide local minima in Fig. 1. Furthermore, we use a feature map visualization that compares methods trained with and without CPR to further corroborate the effectiveness of our method.

## 2 CPR: CLASSIFIER-PROJECTION REGULARIZATION FOR WIDE LOCAL MINIMUM

In this section, we elaborate in detail the core motivation outlined in Fig. 1, then formalize CPR as the combination of two regularization terms: one stemming from prior regularization-based CL methods, and the other that promotes a wide local minima. Moreover, we provide an information-geometric interpretation (Csiszár, 1984; Cover & Thomas, 2012; Murphy, 2012) for the observed gain in performance when applying CPR to CL.

We consider continual learning of $T$ classification tasks, where each task contains $N$ training sample-label pairs $\{(\mathbf{x}_n^t, y_n^t)\}_{n=1}^N$, $t \in [1, \cdots, T]$ with $\mathbf{x}_n^t \in \mathbb{R}^d$, and the labels of each task has $M_t$ classes, *i.e.*, $y_n^t \in [1, \cdots, M_t]$. Note that task boundaries are *given* in evaluation time; i.e., we consider a task-aware setting. We denote $f_{\boldsymbol{\theta}} : \mathbb{R}^d \to \Delta_M$ as a neural network-based classification model with softmax output layer parameterized by $\boldsymbol{\theta}$.

## 2.1 MOTIVATION: INTRODUCING WIDE LOCAL MINIMA IN CONTINUAL LEARNING

Considering the setting of typical regularization-based CL (top of Fig. 1), we denote $\boldsymbol{\theta}_i^*$ as parameters that achieve local minima for a specific task $i$ and $\hat{\boldsymbol{\theta}}_i$ is that obtained with regularization terms. Assuming that $\boldsymbol{\theta}_1^*$ is learnt, when learning task 2, an appropriate regularization updates the parameters from $\boldsymbol{\theta}_1^*$ to $\hat{\boldsymbol{\theta}}_2$ instead of $\boldsymbol{\theta}_2^*$, since $\hat{\boldsymbol{\theta}}_2$ achieves low-errors on both tasks 1 and 2. However, when the low-error regimes (ellipsoids in Fig. 1) are narrow, it is often infeasible to obtain a parameter that performs well on all three tasks. This situation results in the trade-off between stability and plasticity in regularization-based CL (Carpenter & Grossberg, 1987). Namely, stronger regularization strength (direction towards past tasks) brings more stability ($\hat{\theta}_3^1$), and hence less forgetting on past tasks. In contrast, weaker regularization strength (direction towards future tasks) leads to more plasticity so that the updated parameter $\hat{\theta}_3^2$ performs better on recent tasks, at the cost of compromising the performance of past tasks.

A key problem in the previous setting is that the parameter regimes that achieve low error for each task are often narrow and do not overlap with each other. Therefore, a straightforward solution is to *enlarge* the low-error regimes such that they have non-empty intersections with higher chance. This observation motivates us to consider wide local minima for each task in CL (bottom of Fig. 1). With wide local minima for each task, a regularization-based CL can more easily find a parameter, $\hat{\boldsymbol{\theta}}_3$, that is close to the the local minimas for each task, *i.e.*, $\{\boldsymbol{\theta}_i^*\}_{i=1}^3$. Moreover, it suggests that once we promote the wide local minima of neural networks during continual learning, both the stability and plasticity could potentially be improved and result in simultaneously higher accuracy for all task — which is later verified in our experimental (see Sec. 3). In the next section, we introduce the formulation of wide local minima in CL.

## 2.2 CLASSIFIER PROJECTION REGULARIZATION FOR CONTINUAL LEARNING

**Regularization-based continual learning**    Typical regularization-based CL methods attach a regularization term that penalizes the deviation of important parameters learned from past tasks in order to mitigate catastrophic forgetting. The general loss form for these methods when learning task $t$ is

$$L_{\mathsf{CL}}^t(\boldsymbol{\theta}) = L_{\mathsf{CE}}^t(\boldsymbol{\theta}) + \lambda \sum_i \Omega_i^{t-1}(\theta_i - \theta_i^{t-1})^2, \tag{1}$$

where $L_{\mathsf{CE}}^t(\boldsymbol{\theta})$ is the ordinary cross-entropy loss function for task $t$, $\lambda$ is the dimensionless regularization strength, $\boldsymbol{\Omega}^{t-1} = \{\Omega_i^{t-1}\}$ is the set of estimates of the weight importance, and $\{\theta_i^{t-1}\}$ is the parameter learned until task $t-1$. A variety of previous work, *e.g.*, EWC (Kirkpatrick et al., 2017), SI (Zenke et al., 2017), MAS (Aljundi et al., 2018), and RWalk (Chaudhry et al., 2018), proposed different ways of calculating $\boldsymbol{\Omega}^{t-1}$ to measure weight importance.

**Single-task wide local minima**    Several recent schemes have been proposed (Pereyra et al., 2017; Szegedy et al., 2016; Zhang et al., 2018) to promote wide local minima of a neural network for solving a single task. These approaches can be unified by the following common loss form

$$L_{\mathsf{WLM}}(\boldsymbol{\theta}) = L_{\mathsf{CE}}(\boldsymbol{\theta}) + \frac{\beta}{N} \sum_{n=1}^N D_{\mathsf{KL}}(f_{\boldsymbol{\theta}}(\mathbf{x}_n) \| g), \tag{2}$$

where $g$ is some probability distribution in $\Delta_M$ that regularizes the classifier output $f_{\boldsymbol{\theta}}$, $\beta$ is a trade-off parameter, and $D_{\mathsf{KL}}(\cdot \| \cdot)$ is the KL divergence (Cover & Thomas, 2012). Note that, for example, when $g$ is uniform distribution $P_U$ in $\Delta_M$, the regularization term corresponds to entropy maximization proposed in Pereyra et al. (2017), and when $g$ is another classifier's output $f_{\boldsymbol{\theta}'}$, Eq. (2) becomes equivalent to the loss function in Zhang et al. (2018).

**CPR: Achieving wide local minima in continual learning**   Combining the above two regularization terms, we propose the CPR as the following loss form for learning task $t$:

$$L_{\mathsf{CPR}}^t(\boldsymbol{\theta}) = L_{\mathsf{CE}}^t(\boldsymbol{\theta}) + \frac{\beta}{N} \sum_{n=1}^{N} D_{\mathsf{KL}}(f_{\boldsymbol{\theta}}(\mathbf{x}_n^t) \| P_U) + \lambda \sum_i \Omega_i^{t-1}(\theta_i - \theta_i^{t-1})^2, \qquad (3)$$

where $\lambda$ and $\beta$ are the regularization parameters. The first regularization term promotes the wide local minima while learning task $t$ by using $P_U$ as the regularizing distribution $g$ in (2), and the second term is from the typical regularization-based CL. Note that this formulation is oblivious to $\boldsymbol{\Omega}_{t-1}$ and, hence, it can be applied to *any* state-of-the-art regularization-based CL methods. In our experiments, we show that the simple addition of the KL-term can significantly boost the performance of several representative state-of-the-art methods, confirming our intuition on wide local minima for CL given in Section 2.1 and Fig 1. Furthermore, we show in the next section that the KL-term can be geometrically interpreted in terms of information projections (Csiszár, 1984; Cover & Thomas, 2012; Murphy, 2012), providing an additional argument (besides promoting wide local minima) for the benefit of using CPR in continual learning.

## 2.3   INTERPRETATION BY INFORMATION PROJECTION

Minimizing the KL divergence terms in (2) and (3) can be expressed as the optimization $\min_{Q \in \mathcal{Q}} D_{\mathsf{KL}}(Q\|P)$, where $P$ is a given distribution and $\mathcal{Q}$ is a convex set of distributions in the probability simplex $\Delta_m \triangleq \{\mathbf{p} \in [0,1]^m | \sum_{i=1}^m \mathbf{p}_i = 1\}$. In other words, the optimizer $P^*$ is a distribution in $\mathcal{Q}$ that is "closest" to $P$, where the distance is measured by the KL divergence, and is termed the information projection (also called I-projection), i.e.,

$$P^* = \arg \min_{Q \in \mathcal{Q}} D_{\mathsf{KL}}(Q\|P). \qquad (4)$$

The quantity $P^*$ has several operational interpretations in information theory (e.g., in universal source coding (Cover & Thomas, 2012)). In addition, the information projection enables a "geometric" interpretation of KL divergence, where $D_{\mathsf{KL}}(Q\|P)$ behaves as the squared Euclidean distance, and $(Q, P^*, P)$ form a "right triangle". The following lemma resembles the Pythagoras' triangle inequality theorem (not satisfied in general by the KL divergence) in information geometry (Cover & Thomas, 2012).

**Lemma 1.** *Suppose $\exists P^* \in \mathcal{Q}$ such that $D_{\mathsf{KL}}(P^*\|P) = \min_{Q \in \mathcal{Q}} D_{\mathsf{KL}}(Q\|P)$, then*

$$D_{\mathsf{KL}}(Q\|P) \geq D_{\mathsf{KL}}(Q\|P^*) + D_{\mathsf{KL}}(P^*\|P), \ \forall Q \in \mathcal{Q}. \qquad (5)$$

In this sense, when minimizing the KL divergence terms in (2) and (3), we project the classifier output $f_{\boldsymbol{\theta}}(\cdot)$ towards a given distribution (e.g., a uniform distribution). Since $f_{\boldsymbol{\theta}}(\cdot)$ can be viewed as a conditional probability distribution $Q_{Y|X}$, where $Y$ is the class label and $X$ is the input, we consider an extension of the information projection to conditional distributions. We call this extension as the *classifier projection*, which seeks a classifier output $P_{Y|X}^*$ in a set $\mathcal{C}$ that is closest (measured by the expected KL divergence) to a given conditional distribution $P_{Y|X}$. Formally, given a convex set $\mathcal{C}$ of conditional distributions, the classifier projection is defined as

$$P_{Y|X}^* = \arg \min_{Q_{Y|X} \in \mathcal{C}} \mathbb{E}_{P_X} \left[ D_{\mathsf{KL}}(Q_{Y|X}(\cdot|X) \| P_{Y|X}(\cdot|X)) \right]. \qquad (6)$$

The KL divergence terms in (2) and (3) are exactly the empirical version of the objective in (6) by taking $Q_{Y|X}(\cdot|X) = f_{\boldsymbol{\theta}}(\cdot)$ and $P_{Y|X}(\cdot|X) = g$ or $P_U$.

Now, we explain why classifier projection can help overcoming catastrophic forgetting. Let the classifier after training task 1 be $P_{Y|X}^{1*} \in \mathcal{C} \subset \Delta_m$, and suppose we have two classifiers for task 2, one is $P_{Y|X}^2 \notin \mathcal{C}$ and the other is $P_{Y|X}^{2*}$ by projecting $P_{Y|X}^2$ to $\mathcal{C}$ using (6), the triplet $(P_{Y|X}^{1*}, P_{Y|X}^2, P_{Y|X}^{2*})$ forms a right triangle, and $P_{Y|X}^{1*}$ and $P_{Y|X}^{2*}$ are closer to each other measured by the KL divergence (Lemma 1). Therefore, when evaluating on task 1, the classifier of task 2 after projection is more similar to each other in terms of cross-entropy (transformed from the KL divergence), guaranteeing a smaller change in training loss and accuracy. The following proposition formally summarizes this paragraph.

**Proposition 1.** *For any classifier $P_{Y|X}^{t-1*} \in \mathcal{C}$ for task $t-1$ with data distribution $P_X^{t-1}$, and let any classifier for task $t$ be $P_{Y|X}^t \notin \mathcal{C}$ and $P_{Y|X}^{t*}$ be the projected classifier by (6), then*

$$\mathbb{E}_{P_{Y|X}^{t-1*} P_X^{t-1}} \left[ -\log P_{Y|X}^t P_X^{t-1} \right] \geq \mathbb{E}_{P_{Y|X}^{t-1*} P_X^{t-1}} \left[ -\log P_{Y|X}^{t*} P_X^{t-1} \right]. \tag{7}$$

To implement the CPR, we need to pre-define the set of possible classifiers $\mathcal{C}$ for projection. An intuitively best choice is the set of classifiers that perform well on all tasks (can be obtained by training all tasks simultaneously). However, in CL setting, such classifiers are not available; therefore, we pick the set of possible classifiers $\mathcal{C}$ to be a KL divergence ball centered at the uniform distribution $P_U$, *i.e.*,

$$\mathcal{C}(P_U, \epsilon) \triangleq \{Q_{Y|X} \in \Delta_M \mid \mathbb{E}_X \left[ D_{KL}(Q_{Y|X} \| P_U) \right] \leq \epsilon\}.$$

We select $P_U$ since it is the centroid of $\Delta_M$ and, hence, the worst-case divergence between any distribution and $P_U$ is at most $\log M$. From the vantage point of classifier projection, the CPR regularization term in (3) can be viewed as the Lagrange dual of the constraint $Q_{Y|X} \in \mathcal{C}(P_U, \epsilon)$—the term that projects the classifier of individual tasks towards the uniform distribution in order to minimize changes when training sequential tasks (See Fig. 2).

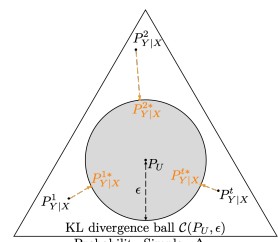

Figure 2: CPR can be understood as a projection onto a finite radius ball around $P_U$.

## 3 EXPERIMENTAL RESULTS

We apply CPR to four regularization-based supervised CL methods: EWC (Kirkpatrick et al., 2017), SI (Zenke et al., 2017), MAS (Aljundi et al., 2018), RWalk (Chaudhry et al., 2018) and AGS-CL (Jung et al., 2020), and further analyze CPR via ablation studies and feature map visualizations.

### 3.1 DATA AND EVALUATION METRICS

We select CIFAR-100, CIFAR-10/100 (Krizhevsky et al., 2009), Omniglot (Lake et al., 2015), and CUB200 (Welinder et al., 2010) as benchmark datasets. Note that we ignore the permuted-MNIST dataset (LeCun et al., 1998) since most state-of-the-art algorithms can already achieve near perfect accuracy on it. CIFAR-100 is divided into 10 tasks where each task has 10 classes. CIFAR-10/100 additionally uses CIFAR-10 for pre-training before learning tasks from CIFAR-100. Omniglot has 50 tasks, where each task is a binary image classification on a given alphabet. For these datasets, we used a simple feed-forward convolutional neural network (CNN) architecture. For the more challenging CUB200 dataset, which has 10 tasks with 20 classes for each task, we used a pre-trained ResNet-18 (He et al., 2016) as the initial model. Training details, model architectures, hyperparameters tuning, and source codes are available in the Supplementary Material (SM).

For evaluation, we first let $a_{k,j} \in [0, 1]$ be the $j$-th task accuracy after training the $k$-th task ($j \leq k$). Then, we used the following three metrics to measure the continual learning performance:

- **Average Accuracy (A)** is the average accuracy $A_k$ on the first $k$ tasks after training the $k$-th task, *i.e.*, $A_k = \frac{1}{k} \sum_{j=1}^k a_{k,j}$. While being a natural metric, Average Accuracy fails to explicitly measure the plasticity and stability of a CL method.
- **Forgetting Measure (F)** evaluates stability. Namely, we define the forgetting measure $f_k^j$ of the $j$-th task after training $k$-th task as $f_k^j = \max_{l \in \{j, \ldots, k-1\}} a_{l,j} - a_{k,j}, \forall j < k$, and the *average forgetting measure* $F_k$ of a CL method as $F_k = \frac{1}{k-1} \sum_{j=1}^{k-1} f_k^j$.
- **Intransigence Measure (I)** measures the plasticity. Let $a_j^\star$ be accuracy of a model trained by fine-tuning for the $j$-the task *without* applying any regularization (in other words, only task-specific loss is used). The intransigence measure $I_{s,k}$ is then defined as $I_{s,k} = \frac{1}{k-s+1} \sum_{j=s}^k i_j$, where $i_j = a_j^\star - a_{j,j}$.

The $F$ and $I$ metrics were originally proposed in (Chaudhry et al., 2018), and we slightly modified their definitions for our usage. Note that a **low** $F_k$ and $I_{1,k}$ implies high stability (low forgetting) and high plasticity (good forward transfer) of a CL method, respectively.

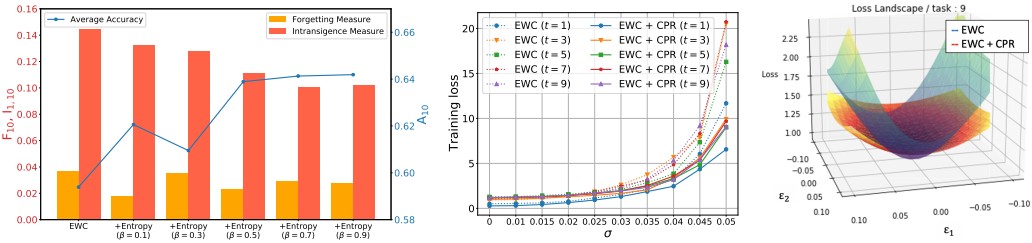

(a) Selecting regularization strength for CPR     (b) Adding Gaussian noise     (c) Analysis using PyHessian

Figure 3: Verifying the regularization for wide local minima

## 3.2 QUANTIFYING THE ROLE OF WIDE LOCAL MINIMA REGULARIZATION

We first demonstrate the effect of applying CPR with varying trade-off parameter $\beta$ in (3) by taking EWC (Kirkpatrick et al., 2017) trained on CIFAR-100 as a running example. Fig. 3(a) shows how the aforementioned metrics varies as $\beta$ changes over $[0.1, \ldots, 1]$. First, we observe that $A_{10}$ certainly increases as $\beta$ increases. Moreover, we can break down the gain in terms of $I_{1,10}$ and $F_{10}$; we observe $I_{1,10}$ monotonically decreases as $\beta$ increases, but $F_{10}$ does not show the similar monotonicity although it also certainly decreases with $\beta$. This suggests that enlarged wide local minima is indeed helpful for improving both plasticity and stability. In the subsequent experiments, we selected $\beta$ using validation sets by considering all three metrics; among the $\beta$'s that achieve sufficiently high $A_{10}$, we chose one that can reduce $F_{10}$ more than reducing $I_{1,10}$, since it turns out improving the stability seems more challenging. (In fact, in some experiments, when we simply consider $A_{10}$, the chosen $\beta$ will result in the lowest $I_{1,10}$ but with even higher $F_{10}$ than the case without CPR.) For comparison purposes, we also provide experiments using Deep Mutual Learning (Zhang et al., 2018) and Label Smoothing (Szegedy et al., 2016) regularizer for achieving the wide local minima in the SM; their performance was slightly worse than CPR.

With the best $\beta$ in hand, Fig. 3(b) experimentally verifies whether using CPR indeed makes the local minima wide. Following the methodology in Zhang et al. (2018), we perturb the network parameters, after learning the final task, of EWC and EWC+CPR by adding Gaussian noise with increasing $\sigma$, then measure the increase in *test* loss for each task (for CIFAR-100). From the figure, we clearly observe that EWC+CPR has a smoother increase in test loss compared with EWC (without CPR) in each task. This result empirically confirms that CPR indeed promotes wide local minima for each task in CL settings and validates our initial intuition given in Sec. 2.1. In the SM, we repeat the same experiment with MAS (Aljundi et al., 2018).

To plot the loss landscape of each model directly, we used PyHessian (Yao et al., 2019), which is the framework that can plot the loss landscape of NNs by perturbing the model parameters across the first and second Hessian eigenvectors. As an example, Figure 3(c) plots and compares the loss landscapes on the training data of the 2nd task of CIFAR-100, for the networks trained with EWC and EWC+CPR, respectively. It clearly shows that, by applying CPR, the loss landscape becomes much wider than that of the vanilla EWC. We consistently observe such trend across all tasks, of which are visualized in the SM, and we believe this confirms our intuition for adding the CPR term.

## 3.3 COMPARISON WITH STATE-OF-THE-ART

Next, we apply CPR to the state-of-the-art regularization-based CL on the benchmark datasets and measure the performance improvement with the three metrics in Section 3.1. For the regularization strengths, we first select the best $\lambda$ without CPR, then choose $\beta$ according to the procedure in Section 3.2. The results in Table 1 are averaged over 10 repeated experiments with different random initialization and task sequence using the chosen $(\lambda, \beta)$. The hyperparameters are reported in the SM.

**CIFAR-100 and CIFAR-10/100** In Table 1 and Fig. 4(a), we observe that CPR consistently improves *all* regularization-based methods for *all* tested datasets in terms of increasing $A_{10}$ and decreasing $I_{1,10}$, and also consistently decreases $F_{10}$. Additionally, we find that for CIFAR-10/100, the orders of the 10 tasks in CIFAR-100 and CIFAR-10 affect the performance of the CPR; namely, in the SM, we show that when CIFAR-10 tasks are positioned in different positions rather than at the beginning, the gain due to CPR got much bigger.

Table 1: Experimental results on CL benchmark dataset with and without CPR.

| Dataset | Method | Average Accuracy ($A_{10}$) | | | Forgetting Measure ($F_{10}$) | | | Intransigence Measure ($I_{1,10}$) | | |
|---|---|---|---|---|---|---|---|---|---|---|
| | | W/o CPR | W/ CPR | diff (W-W/o) | W/o CPR | W/ CPR | diff (W/-W/o) | W/o CPR | W/ CPR | diff (W-W/o) |
| CIFAR100 ($T=10$) | EWC | 0.6002 | 0.6328 | **+0.0326 (+5.2%)** | 0.0312 | 0.0285 | **-0.0027 (-8.7%)** | 0.1419 | 0.1117 | **-0.0302 (-21.3%)** |
| | SI | 0.6141 | 0.6476 | **+0.0336 (+5.5%)** | 0.1106 | 0.0999 | **-0.0107 (-9.7%)** | 0.0566 | 0.0327 | **-0.0239 (-42.2%)** |
| | MAS | 0.6172 | 0.6442 | **+0.0270 (+4.4%)** | 0.0416 | 0.0460 | **-0.0011 (-2.6%)** | 0.1155 | 0.0778 | **-0.0257 (-22.2%)** |
| | Rwalk | 0.5784 | 0.6366 | **+0.0581 (+10.0%)** | 0.0937 | 0.0769 | **-0.0169 (-18.0%)** | 0.1074 | 0.0644 | **-0.0430 (-40.0%)** |
| | AGS-CL | 0.6369 | 0.6615 | **+0.0246 (+3.9%)** | 0.0259 | 0.0247 | **-0.0012 (-4.63%)** | 0.1100 | 0.0865 | **-0.0235 (-24.4%)** |
| CIFAR10/100 ($T=11$) | EWC | 0.6950 | 0.7055 | **+0.0105 (+1.5%)** | 0.0228 | 0.0181 | **-0.0048 (-21.1%)** | 0.1121 | 0.1058 | **-0.0062 (-5.5%)** |
| | SI | 0.7127 | 0.7186 | **+0.0059 (+0.8%)** | 0.0459 | 0.0408 | **-0.0051 (-11.1%)** | 0.0733 | 0.0721 | **-0.0012 (-1.6%)** |
| | MAS | 0.7239 | 0.7257 | **+0.0017 (+0.2%)** | 0.0479 | 0.0476 | **-0.0003 (-0.6%)** | 0.0603 | 0.0588 | **-0.0015 (-2.5%)** |
| | Rwalk | 0.6934 | 0.7046 | **+0.0112 (+1.6%)** | 0.0738 | 0.0707 | **-0.0031 (-4.2%)** | 0.0672 | 0.0589 | **-0.0084 (-12.5%)** |
| | AGS-CL | 0.7580 | 0.7613 | **+0.0032 (+0.4%)** | 0.0009 | 0.0009 | **0** | 0.0731 | 0.0697 | **-0.0034 (-4.7%)** |
| Omniglot ($T=50$) | EWC | 0.6632 | 0.8387 | **+0.1755 (+26.5%)** | 0.2096 | 0.0321 | **-0.1776 (-84.7%)** | -0.0227 | -0.0239 | **-0.0012 (-5.3%)** |
| | SI | 0.8478 | 0.8621 | **+0.0143 (+1.7%)** | 0.0247 | 0.0167 | **-0.0079 (-32.0%)** | -0.0258 | -0.0282 | **-0.0065 (-25.3%)** |
| | MAS | 0.8401 | 0.8679 | **+0.0278 (+3.3%)** | 0.0316 | 0.0101 | **-0.0215 (-68.0%)** | -0.0247 | -0.0314 | **-0.0067 (-27.1%)** |
| | Rwalk | 0.8056 | 0.8497 | **+0.0440 (+5.5%)** | 0.0644 | 0.0264 | **-0.0380 (-59.0%)** | -0.0226 | -0.0294 | **-0.0068 (-30.1%)** |
| | AGS-CL | 0.8553 | 0.8805 | **+0.0253 (+3.0%)** | 0 | 0 | **0** | 0.0323 | 0.0046 | **-0.0277 (-85.8%)** |
| CUB200 ($T=10$) | EWC | 0.5746 | 0.6098 | **+0.0348 (+6.1%)** | 0.0811 | 0.0807 | **-0.0004 (-0.5%)** | 0.1011 | 0.0667 | **-0.0345 (-34.1%)** |
| | SI | 0.6047 | 0.6232 | **+0.0157 (+2.6%)** | 0.0549 | 0.0474 | **-0.0075 (-13.7%)** | 0.0918 | 0.0827 | **-0.0091 (-9.9%)** |
| | MAS | 0.5842 | 0.6123 | **+0.0281 (+4.8%)** | 0.1188 | 0.1030 | **-0.0158 (-13.3%)** | 0.0575 | 0.0436 | **-0.0139 (-24.2%)** |
| | Rwalk | 0.6078 | 0.6324 | **+0.0247 (+4.1%)** | 0.0811 | 0.0601 | **-0.0210 (-25.9%)** | 0.0679 | 0.0621 | **-0.0058 (-8.5%)** |
| | AGS-CL | 0.5403 | 0.5623 | **+0.0220 (+4.07%)** | 0.0750 | 0.0692 | **-0.0058 (-7.7%)** | 0.1408 | 0.1241 | **-0.0167 (-11.7%)** |

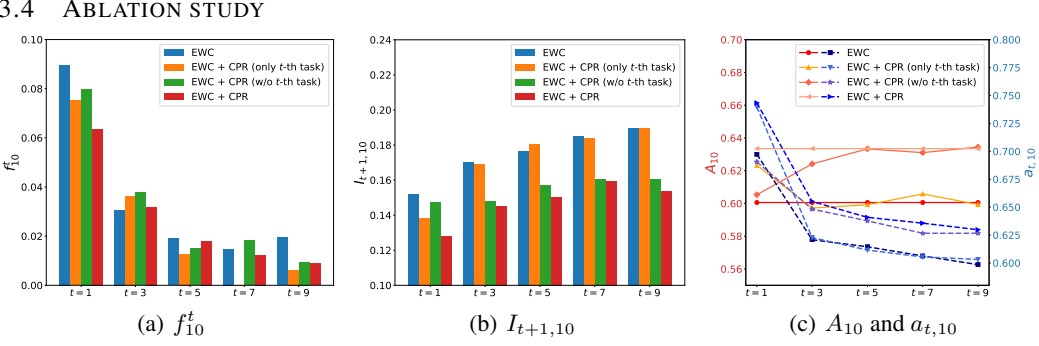

| (a) CIFAR-100 | (b) Omniglot | (c) CUB200 |
|---|---|---|

Figure 4: Experimental results on CL benchmark dataset

**Omniglot** This dataset is well-suited to evaluate CL with long task sequences (50 tasks). In Table 1, it is clear that the CPR considerably increases both plasticity and stability in long task sequences. In particular, CPR significantly decreases $F_{10}$ for EWC and leads to a huge improvement in $A_{10}$. Interestingly, unlike the previous datasets, $I_{1,10}$ is *negative*, implying that past tasks help in learning new tasks for the Omniglot dataset; when applying CPR, the gains in $I_{1,10}$ are even better. Furthermore, Fig. 4(b) indicates that applying CPR leads to less variation in $A_t$ curves.

**CUB200** The results in Table 1 and Fig. 4(c) show that CPR is also effective when using a pre-trained ResNet model for all methods and metrics.

### 3.4 ABLATION STUDY

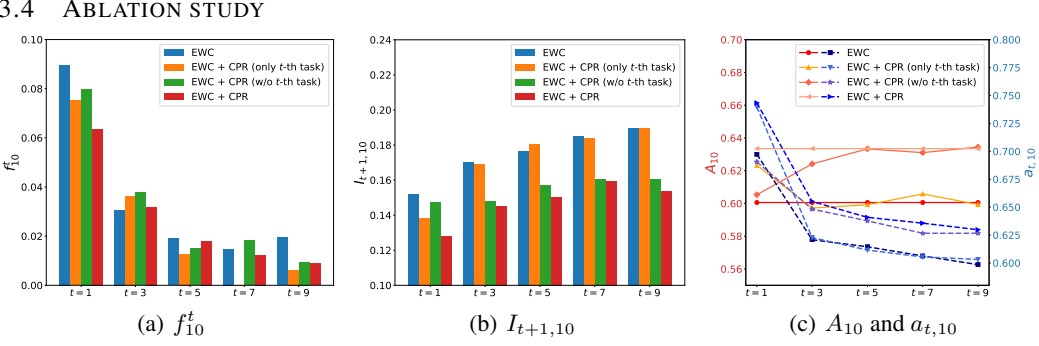

| (a) $f_{10}^t$ | (b) $I_{t+1,10}$ | (c) $A_{10}$ and $a_{t,10}$ |
|---|---|---|

Figure 5: Ablation studies on CL with wide local minima

We study the ablation of the CPR on the regularization-based methods using CIFAR-100 with the best ($\lambda$, $\beta$) found previously, and report the averaged results over 5 random initializations and task sequences in Fig. 5. The ablation is performed in two cases: (i) using CPR only at task $t$, denoted as EWC + CPR (only $t$-th task), and (ii) using CPR except task $t$, denoted as EWC + CPR (w/o $t$-th task). Fig. 5(a) shows $f_{10}^t$, the amount of forgetting for task $t$ after learning the task 10, and Fig. 5(b) shows

$I_{t+1,10}$, the amount of gap with fine-tuning after task $t$. In Fig. 5(a), we observe that CPR helps to decrease $f_{10}^t$ for each task whenever it is used (except for task 3), but $f_{10}^t$ of EWC + CPR (w/o $t$-th task) shows a more random tendency. On average, EWC + CPR does reduce forgetting in all tasks, demonstrating the effectiveness of applying CPR to all tasks. Notably in Fig. 5(b), $I_{t+1,10}$ of EWC + CPR (only $t$-th task) is lower than that of EWC + CPR (w/o $t$-th task) only when $t = 1$; this indicates that CPR is most beneficial in terms of plasticity when CPR is applied as early as possible to the learning sequence. EWC + CPR again achieves the lowest (i.e., most favorable) $I_{t+1,10}$. Fig. 5(c), as a further evidence, also suggests that applying CPR for $t = 1$ gives a better accuracy. Moreover, the accuracy of EWC + CPR (w/o $t$-th task) gets closer to the optimal EWC + CPR, which is consistent with the decreasing difference of $I_{t+1,10}$ between EWC + CPR (w/o $t$-th task) and EWC + CPR in Fig. 5(b). The EWC + CPR still gives the best $A_{10}$ and individual $a_{t,10}$ accuracy. We emphasize that model converging to a wide local minima from the first task onwards considerably helps the training of future tasks as well, *i.e.*, a significant increase in the plasticity can be achieved. By using this finding, we conducted an experiment on the case where CPR have to learn unscheduled additional tasks and got the impressive experimental result which is reported in the SM.

In the SM, we also visualize how the learned representations change with the additional CPR term. We utilized UMAP (McInnes et al., 2018) and show that the learned representations less drift as the learning continues with CPR, which again indirectly corroborates the existence of wide local minima. Due to the space constraint, we refer the detailed explanation and visualization to the SM.

### 3.5 Applying CPR to Continual Reinforcement Learning

To show the effectiveness of CPR in various domain, we applied CPR to EWC (Kirkpatrick et al., 2017), MAS (Aljundi et al., 2018) and AGS-CL (Jung et al., 2020) in continual RL. We followed exactly same experimental settings with AGS-CL, therefore, we did experiments on 8 different Atari (Brockman et al., 2016) tasks, *i.e.*, $\{StarGunner - Boxing - VideoPinball - Crazyclimber - Gopher - Robotank - DemonAttack - NameThisGame\}$. We used PPO (Schulman et al., 2017) for reinforcement learning and we simply applied CPR to PPO by adding KL divergence term in Eq.2. We trained each method on three different seeds and we report the averaged result. More detailed experimental settings are in the SM.

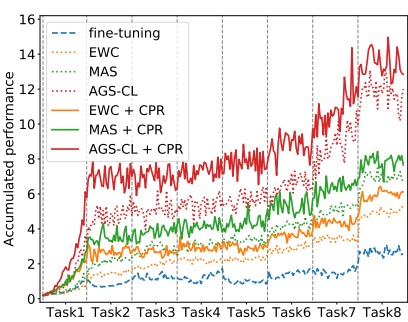

Figure 6: Accumulated normalized reward

Figure 6 shows the experimental results and fine-tuning means the result of continual learning without regularization. x-axis is the training step and y-axis means the normalized accumulated reward, which is the sum of each task reward normalized by the reward of fine-tuning. We observe that CPR increases the accumulated reward of each method. From the analysis, we found out that, training with the best hyperparameter for each method already do not suffer from catastrophic forgetting on previous tasks but each method shows the difference on the ability to learn a new task well. However, we check that CPR can averagely increase the average reward of each task by 27%(EWC), 6%(MAS) and 5%(AGS-CL) and we believe that this is the reason why CPR leads to the improvement of the accumulated reward of each method.

## 4 Related Work

Several methods have been recently proposed to reduce catastrophic forgetting (see Parisi et al. (2019) for a survey). In this paper, we mainly focus on the regularization-based CL methods (Li & Hoiem, 2017; Kirkpatrick et al., 2017; Aljundi et al., 2018; Chaudhry et al., 2018; Zenke et al., 2017; Nguyen et al., 2018; Ahn et al., 2019; Aljundi et al., 2019; Jung et al., 2020). Broadly speaking, the motivation behind the regularization-based CL is to measure the importance of model parameters in previous tasks. This measure is then used in a regularization term for overcoming catastrophic forgetting when training for new tasks. Consequently, the main research focus of the regularization-based CL is creating metrics for quantifying weight importance on previous tasks (e.g., Kirkpatrick et al. (2017); Aljundi et al. (2018); Chaudhry et al. (2018); Zenke et al. (2017); Nguyen et al. (2018); Ahn et al.

(2019); Jung et al. (2020)). In contrast, here we focus on developing a general method for augmenting regularization-based CL instead of proposing (yet another) new metric for measuring the weight importance.

The work that shares similar philosophy as ours is Aljundi et al. (2019), which encourages sparsity of representations for each task by adding an additional regularizer to the regularization-based CL methods. Note that the motivation of Aljundi et al. (2019)—imposing sparsity of neuron activations— is considerably different from ours that promotes wide local minima. Moreover, whereas Aljundi et al. (2019) focuses on average accuracy, we carefully evaluate the advantage of the added CPR regularization in terms of increasing both plasticity and stability of CL *in addition to* accuracy.

Several papers have recently proposed methods that promote wide local minima in neural networks in order to improve single-task generalization, including using small mini-batch size (Keskar et al., 2017), regularizing the output of the softmax layer in neural networks (Szegedy et al., 2016; Pereyra et al., 2017), using a newly proposed optimizer which constructs a local-entropy-based objective function (Pereyra et al., 2017) and distilling knowledge from other models (Zhang et al., 2018). We expand upon this prior work and investigate here the role of wide local minima in CL.

Mirzadeh et al. (2020) recently proposed a similar point of view with ours on the advantages of wide local minina in continual learning. However, our work is significantly different from theirs in the following aspects. Firstly, the core motivations are different. Mirzadeh et al. (2020) begins with defining a metric for forgetting, then approximates it with the second order Taylor expansion. They then mainly focus on the stability and argue that, if the model converges to a wide local minima during continual learning, the forgetting would decrease. However, as shown in Fig. 1, our paper is motivated from a geometric intuition, from which we further explain that if the model achieves a wide local minima for each task during continual learning, not only stability but also plasticity will improve (in other words, the trade-off between stability and plasticity in continual learning can be improved simultaneously). Secondly, the proposed method for converging to a wide local minima is different. Mirzadeh et al. (2020) proposed to control three elements, such as, learning rate, mini-batch size and drop out. On the other hand, we used classifier projection as a regularization that promotes wide local minima. Therefore, our method requires only one additional hyperparameter to be controlled, so the complexity of our method is much lower. Thirdly, while Mirzadeh et al. (2020) only empirically analyzed the forgetting of CL, we proposed a more principled theoretical interpretation of the role of CPR in terms of information projection. Fourthly, unlike Mirzadeh et al. (2020) which only considered a single epoch setting on a limited benchmark sets, we conducted extensive experiments on multiple epochs setting and diverse settings, such as, four different classification datasets (CIFAR100, CIFAR10/100, Omniglot, CUB200) and continual reinforcement learning using Atari 8 tasks. Finally, there is a difference in experimental analyses. We conducted a more detailed experimental analysis of the effect of wide local minima for continual learning. This is because, by using UMAP, we verified the change of feature maps of a model with or without CPR and how the plasticity and stability were improved after applying CPR through ablation study.

## 5 Concluding Remark

We proposed a simple classifier-projection regularization (CPR) which can be combined with *any* regularization-based continual learning (CL) method. Through extensive experiments in supervised and reinforcement learning, we demonstrated that, by converging to a wide local minima at each task, CPR can significantly increase the plasticity and stability of CL. These encouraging results indicate that wide local minima-promoting regularizers have a critical role in successful CL. As a theoretical interpretation, we argue that the additional term found in CPR can be understood as a projection of the conditional probability given by a classifier's output onto a ball centered around the uniform distribution.

## Acknowledgment

This work was supported in part by NRF Mid-Career Research Program [NRF-2021R1A2C2007884] and IITP grant [No.2019- 0-01396, Development of framework for analyzing, detecting, mitigating of bias in AI model and training data], funded by the Korean government (MSIT).

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
