# OpenReview forum: "CPR: Classifier-Projection Regularization for Continual Learning"
_ICLR.cc/2021/Conference — ICLR 2021 Poster_

### Official Review · AnonReviewer3 · 2020-10-19
**Finding wide local minima to help continual learning**

**Rating:** 7
**Confidence:** 4

**Review:**

The authors make the intuitive argument that finding wide local minima in DNN training is particularly desirable for Continual Learning and thus propose a simple loss-modification that is applicable out-of-the-box to a wide array of CL methods. The intuitive argument is supported by a good experimental analysis and a theoretical argument.

Pros:
- The method is trivial to implement and thus very general.
- The authors show very impressive improvements across three performance measures for various recent algorithms, indicating a benefit in almost all cases. I was pleased to see an application in RL, which is often shunned by most works in CL.
- Experimental analysis is to a very high standard, investigating the source of aforementioned performance improvements with accepted measures of minima depth.
- The paper is overall well written, intuitive and easy to follow

Cons:
- The main idea behind this paper is arguably rather simple, straight-forwardly applying a known technique in a new setting. Nevertheless, the CL community may be ignorant to the particular effectiveness of the technique in this setting, which is why this paper makes for a suitable publication.

---

### Official Review · AnonReviewer2 · 2020-10-23
**CPR review**

**Rating:** 6
**Confidence:** 4

**Review:**

The authors argue that achieving wide local minima during the training of tasks, is beneficial for continual learning. The plausible intuition (explained in Fig. 1) is that it is easier to find a parameter setting that is beneficial for all tasks when tasks have wide local minima. They enforce wide local minima by adding an entropy loss to the classifier ( a known strategy). The loss is further combined with any weight regularization loss, like EWC, SI, MAS.

pros.
- motivation is plausible
- results show that the simple method improves results over a wide range of weight regularization methods.
- the simplicity of the method could mean that will be widely adopted in CL losses.
- experiments also includes results for reinforcement learning

cons:
- novelty is rather low, it is known that wide minima are desirable. Here they show that they are also (or maybe even more) desirable for CL, and show that enforcing them (with a known method) improves results.
- improve discussion/results of other techniques to encourage classifiers to be less certain about predictions.

Remarks:
The entropy loss seems to enforce the classifier to be less certain about its predictions. Other methods that also have this effect could be discussed in more detail. Also, I could not find the promised results Label Smoothiing in Sup. Mat.

Minor remarks:
- would the loss also work when combined with non-weight regularization losses like LwF ?
- I found the ablation study of less interest and could be moved to sup. mat. if needed.
- At the start of section 2, it would be good to mention that methods are evaluated in a task-aware setting (not task agnostic).
- many citations are in the wrong format (the authors should check how to correctly use the cite, citep, citet (I am not sure))

Final recommendation: The proposed method is very simple, but could be widely applied in CL losses and the paper could therefore have an impact.

---

### Official Review · AnonReviewer1 · 2020-10-29
**A simple and heuristic regularization, theoretical gaps, missing comparison on standard benchmarks and other CL methods, related works in CL about flat minima.**

**Rating:** 4
**Confidence:** 4

**Review:**

The paper proposes to add a KL-divergence regularization to the objective of regularized continual learning in order to encourage the output prediction to be close to a uniform distribution over classes (i.e., increasing the entropy). They argue that this regularization makes the local minima flat and thus less prone to forgetting. They try to build a theoretical connection using results from information projection but there is still a large gap. In experiments, they show on several benchmarks that applying the KL divergence regularization to different regularization-based continual learning brings improvements.

It is always interesting to find a simple method that can consistently bring improvements. However, the simple method in this paper lacks sufficient explanation and theoretical justification. The experiments are also less convincing if without comparison to other types of continual learning methods.

(1) The paper uses too much space (~4 pages) to explain the motivation of preferring flat minima and KL divergence and make them overcomplicated, which are actually quite straightforward to understand. However, **it uses less than 1 page for theoretical justification of the main idea (Section 2.3)** with few explanations about the equations and a large gap in building the connection between "flat minima" and KL divergence regularization. In particular, in Proposition 1:

(i) The cross-entropy on the left or right-hand side of Eq.7 is defined on two joint distributions in the form of $P(X,Y)=P(Y|X)P(X)$, i.e., it shows that two joint distributions P1(X,Y) and P2(X,Y) becomes closer after projecting one of them to the KL ball. **However, what we truly care is whether P1(Y|X) and P2(Y|X) are close in expectation of X drawn from P(X), i.e., whether $E_{X~\sim P(X)}[KL(P1|P2)]$ or $E_{X\sim P(X)}[CE(P1|P2)]$ becomes smaller after the projection**, where E is expectation and CE is cross-entropy. To close this gap, the authors need to prove this very specifically.

(ii) The result is mainly due to **an artifact of constraining all previous tasks' classifiers' outputs close to the uniform distribution**. It is obvious that the distance of distribution outside the ball to distribution inside becomes smaller if we project the outside distribution onto the ball. However, previous tasks' classifiers might already perform very poorly on their data distributions because their predictions are too close to uniform sampling, and the KL regularization accumulates the error and only make the prediction quality worse. In the extreme case, one can think that the classifier for task t-1 produces almost uniform predictions so it performs very poorly. Now making the classifier at task t also produce similar uniform predictions for data drawn from task t-1 does not help.

(iii) Proposition 1 holds only when Lemma 1 holds, and Lemma 1 holds only when P* is the minimum solution for minimizing the KL divergence, i.e., Eq.6. However, the KL divergence is only a regularization term in the objective of Eq.3 (and there are two other terms) so it almost cannot be minimized. So **both Lemma 1 and Proposition 1 do not hold rigorously**.

(2) It is encouraging to see that the proposed KL divergence regularization can improve regularization-based continual learning. However, as shown in recent works, other types of continual learning (e.g., memory replay based and dynamic architecture based) usually performs better. Since the proposed regularization can easily work within these methods, it is more convincing to compare their performance before and after adding the regularization.

(3) The experiments do not cover several standard continual learning benchmarks such as permuted/rotated MNIST/FMNIST and sequential MNIST/Tiny-ImageNet.

(4) Encouraging flat minima can alleviate forgetting in continual learning has been analyzed and studied in a recent paper:
Mirzadeh et al., "Understanding the Role of Training Regimes in Continual Learning", NeurIPS 2020. A discussion and comparison to it are recommended. Its analysis is in the model parameter space instead of the output space and is rigorous.

(5) In Figure 3(c), the EWC+CPR achieved flat minima has much worse(larger) loss. Does this indicate that the stability is achieved with the price of significantly degraded loss?

(6) Minor: the notations need to be more rigorous, for example (i) expectations E_{P_X} and E_X are both used in the paper at several positions, but they should be E_{X~P(X)}; (ii) to make the paper and appendix consistent, Q* in the proof of Lemma 1 of the appendix should be P*; (iii) The superscripts and subscripts can be simplified.

---

### Official Review · AnonReviewer4 · 2020-11-08
**This paper proposes an additional regularizer (called classifier projection regularization) to continual training of neural networks that promotes the wideness of the loss landscape and increases the entropy of the predictions. The paper pursues a novel direction in understanding catastrophic forgetting.**

**Rating:** 6
**Confidence:** 5

**Review:**

The viewpoint is quite novel and is built with nice connections to neural network optimization literature.

The paper has claimed several times that one can improve stability and plasticity simultaneously. This somehow intuitively makes less sense because it seems that there is an inherent tradeoff between them. If the model is well calibrated then increasing stability should decrease plasticity and vice versa. That's why it's usually referred to as the stability-plasticity dilemma. It might be the case that improvement in both measures is a sign of a sup-optimal regime.

The paper has also claimed the first one to study the connection between widness of local minima and continual learning. There is at least one paper that has studied this relation before:Mirzadeh, Seyed Iman, et al. "Understanding the role of training regimes in continual learning." arXiv preprint arXiv:2006.06958 (2020).
Since it's been around for a short while the authors were probably not aware of this paper but it basically connects widness of the local minima to stability of continual training. It's worth elaborating on the differences from and similarities to their approach.

The paper is generally well written. But, some notations are a little unclear and can be improved. For example, the difference between \theta_i^start and \hat \theta_i is wauge. Also, citations should be improved. The authors should note the difference between \citet and \citep, etc. A proofreading can improve some typos throughout the paper.

Section 2.3 is not straightforward to follow. For example, it's good to elaborate (e.g. in an appendix section) why proposition 1 is a consequence of lemma 1? Alos, how P_{Y|X}^{t} are related to continual learning steps? The role of P_u is unclear and it's not that obvious where does it appear in equation 7?

Experiments look convincing but it can be improved in terms of clarity of the setup. For example, in measuring intransigence it's mentioned no regularization is applied. It's unclear whether this "regularization" is the proposed CPR (using KL) or the normal EWC-like regularization (like last term in equation 3)? Reporting confidence interval in results could have improved the empirical validation.

---

### Decision · Program_Chairs · 2021-01-07
**Final Decision**

**Decision:**

Accept (Poster)

**Comment:**

There was some positive consensus towards this paper, which slightly improved after the very strong author rebuttal. Reviewers, in general, appreciate the simplicity of the approach as well as its effectiveness. The most acute criticisms derived from several theoretical and technical points, similarity with [Mizadeh, 2020], and missing baseline comparisons. The author rebuttal responds to each of these points very clearly and convincingly, as well as with new experimental baseline comparisons that clearly demonstrate the effectiveness of the CPR approach. I encourage the authors to include the extensive comparison with [Mizadeh, 2020] provided in the rebuttal, especially given the similarity to the proposed approach. and to also tone down the strong claims of novelty in light of the similarities.